# A century of change in the California Current: upwelling system amplifies acidification

Mary Margaret V. Stoll [1] ✉, Curtis A. Deutsch [2], Hana Jurikova [3], James W. B. Rae [3], Hartmut Frenzel[1,4,5], Anne M. Gothmann[6], Simone R. Alin [5] & Alexander C. Gagnon [1]

Predicting the pace of acidification in the California Current System (CCS), a productive upwelling system that borders the west coast of North America, is complex because the anthropogenic contribution is intertwined with other natural sources. A central question is whether acidification in the CCS will follow the pace of increasing atmospheric $CO_2$, or if climate effects and other biogeochemical processes will either amplify or attenuate acidification. Here, we apply the boron isotope pH proxy to cold-water orange cup corals to establish a historic level of acidification in the CCS and the Salish Sea, an associated marginal sea. Through a combination of complementary modeling and geochemical approaches, we show that the CCS and Salish Sea have experienced amplified acidification over the industrial era, driven by the interaction between anthropogenic $CO_2$ and a thermodynamic buffering effect. From this foundation, we project future acidification in the CCS under elevated $CO_2$ emissions. The projected change in $pCO_2$ over the 21st century will continue to outpace atmospheric $CO_2$, posing challenges to marine ecosystems of biological, cultural, and economic importance.

Since the beginning of the Industrial Revolution, the oceans have absorbed approximately a third of anthropogenic $CO_2$ emissions, corresponding to an acidity increase of 25% and a 0.1 decline in pH in the open surface ocean[1,2]. While most open ocean surface waters equilibrate with atmospheric $CO_2$, the magnitude of acidification in coastal regions can be complex due to the combination of both anthropogenic and natural sources of acidity, particularly at depth. For example, in the California Current System (CCS), the eastern boundary current off the west coast of North America (Fig. 1a), intense, episodic coastal upwelling of low-pH, high-$pCO_2$ waters is driven by alongshore winds and associated Ekman transport[3,4]. As a result, the CCS and adjacent waterways, such as the Salish Sea in the Pacific Northwest, represent a leading edge of ocean acidification (OA) impacts while also providing a window into future ocean conditions and processes. Understanding the impact of ocean acidification in this upwelling

system is particularly critical, as the associated delivery of nutrients from the deep ocean fuels one of the most biologically productive and economically vital ecosystems in the world[5]. Even though organisms in the CCS have historically been exposed to acidified and variable conditions, a synthesis of several studies suggests that many regional species are sensitive to OA[6]. Despite its importance and potential vulnerability, the trajectory of future acidification in this upwelling system remains unclear due to the interplay of processes that impact regional biogeochemistry.

A central question is whether acidification in the productive CCS will follow the pace of increasing atmospheric $CO_2$, or if dynamical climate effects or other processes will act to either amplify or attenuate acidification. Processes that can modulate acidification in the CCS include changes in upwelling dynamics, shifts in the source of upwelled water, and changes in the amount of exported material from the

[1]School of Oceanography, University of Washington, Seattle, WA, USA. [2]Department of Geosciences, Princeton University, Princeton, NJ, USA. [3]School of Earth and Environmental Sciences, University of St Andrews, St Andrews, UK. [4]Cooperative Institute for Climate, Ocean, and Ecosystem Studies, University of Washington, Seattle, WA, USA. [5]Pacific Marine Environmental Laboratory, National Oceanic and Atmospheric Administration, Seattle, WA, USA. [6]Departments of Environmental Studies and Physics, St. Olaf College, Northfield, MN, USA. ✉e-mail: mmstoll@uw.edu

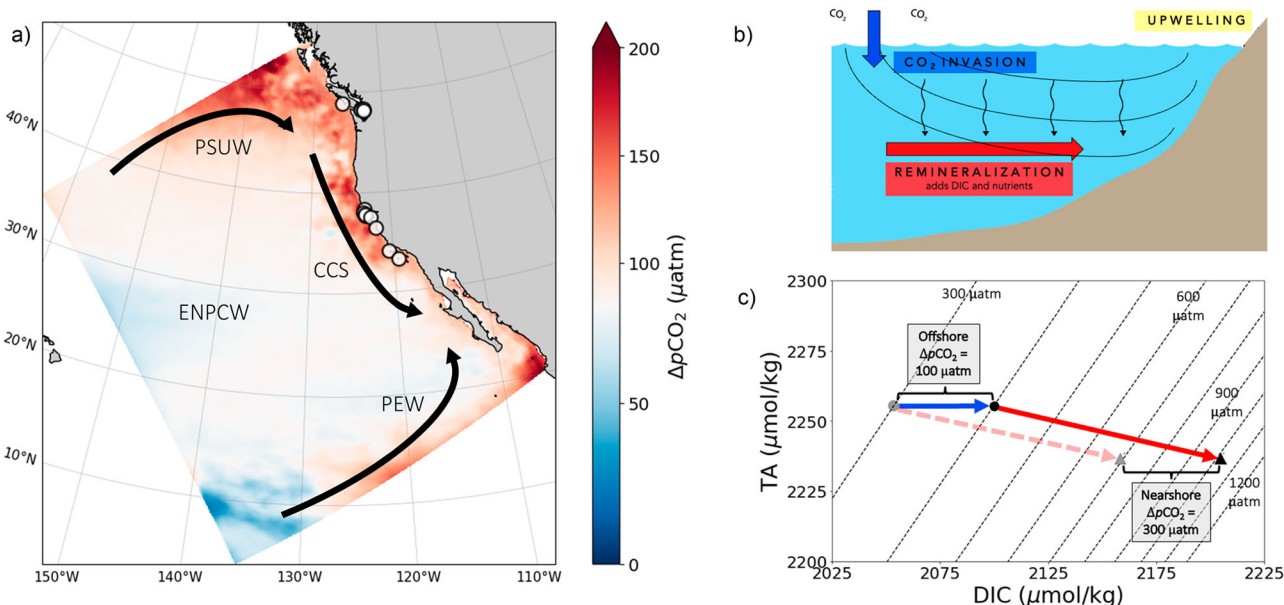

**Fig. 1 | Processes influencing carbonate chemistry in the CCS over the 20th century. a** Climatological mean map of $p$CO$_2$ anomaly at 75 m depth over the ROMS domain. 20th-century change ($\Delta p$CO$_2$) was evaluated by calculating the difference between a 7-year average in the historic (1897–1904) and modern (2000–2007) model simulations. The color bar is centered at 80 µatm to represent the atmospheric change in CO$_2$ over the modeled time period (1900–2000). Red areas in coastal upwelling regions, as well as the northern and southern boundaries, indicate an amplified increase in seawater $p$CO$_2$ relative to the atmosphere. The Pacific Subarctic Upper Water (PSUW), Eastern North Pacific Central Water (ENPCW), and Pacific Equatorial Water (PEW) mix to form the California Current System (CCS), which flows along the US West Coast. Outlined circles represent the historic coral locations. **b** A generalized cartoon section of the California Current upwelling region with a depiction of relevant processes that influence biogeochemistry. Isopycnals outcrop in open ocean locations and exchange with atmospheric CO$_2$ (blue arrow). These outcrops are far from shore, but the figure has been condensed for graphical representation. During transport along an isopycnal from the open ocean, the water mass accumulates additional carbon via remineralization (red arrow) of organic matter exported from the surface ocean (black wavy lines). In the CCS, alongshore winds and associated Ekman transport induce upwelling of deep, carbon-rich waters. **c** The processes presented in (**b**) impact carbonate chemistry in the CCS, depicted by the same color arrows. In the historic era, open ocean outcrops equilibrated with atmospheric CO$_2$ (gray circle), and DIC increased from the open ocean to the CCS due to remineralization (red dashed arrow). In the modern ocean, the offshore outcrop accumulates more carbon (black circle) due to equilibration of anthropogenic CO$_2$ (blue arrow). Even if the magnitude of remineralization remains the same between the historic and modern oceans, and $p$CO$_2$ of the outcrop region exactly follows atmospheric changes, the CCS will experience amplified $\Delta p$CO$_2$ relative to the atmosphere and open ocean region. This is because increased initial DIC in the modern outcrop region results in lower buffering capacity[56] such that the impact of subsequent remineralization on $p$CO$_2$ is amplified. This thermodynamic buffering effect can be seen as tighter contours between lines of constant $p$CO$_2$ as DIC increases. Although all processes depicted in (**b**) play important roles in describing the biogeochemistry of the CCS, we find that anthropogenic carbon and this thermodynamic buffering effect drive the 20th-century change.

surface ocean that is remineralized along upwelled isopycnals (Fig. 1b). Recent research has also brought attention to thermodynamic buffering effects that amplify the impact of this remineralization on seawater pH and $p$CO$_2$ when water masses accumulate anthropogenic carbon (Fig. 1c). This thermodynamic effect highlights the vulnerability to acidification in carbon-rich waters due to low buffering capacity.

Unfortunately, current studies using coupled ocean-atmosphere models and wind stress analyses differ on the predicted sign and magnitude of changes to the mean state and variability of the CCS in the past and future[7–18]. Furthermore, while the effects of thermodynamic buffering have long been acknowledged and attempts have been made to calculate the impact of changing oceanic carbon buffering capacity[19–21], their implications have only recently been applied to evaluate spatial patterns and rates of OA[22–25]. Given the wide range of results in the literature, the role of climate effects and their interplay with biogeochemistry in a changing CCS remains uncertain. Testing these predictions and resolving long-term trends from natural decadal variability requires observational data that extends beyond the limited range of instrumental records to span longer timescales.

To address this knowledge gap, we reconstruct carbonate chemistry along the US West Coast and Salish Sea over the industrial era (1890–2020) through measurements of boron isotopes ($\delta^{11}$B) in the preserved aragonite (CaCO$_3$) skeletons of *Balanophyllia elegans*, a common cold-water coral ("Methods" section). To quantify the magnitude of acidification over the past century, we compare historic museum specimens that grew during the Industrial Revolution (1890s) with modern (2020) corals collected from the same locations ("Methods" section, Supplementary Fig. 1). To attribute the geochemical signals to oceanographic processes and predict future conditions, we contextualize the coral record with a biogeochemical regional ocean modeling system (ROMS). We compare both geochemical and modeled acidification signals to a baseline level of acidification, which we define as the magnitude of acidification associated with the uptake of atmospheric CO$_2$ over the industrial era, to assess whether coastal processes attenuate or amplify acidification from CO$_2$ invasion. We analyze acidification in terms of $p$CO$_2$ as opposed to pH, allowing us to directly compare magnitudes of acidification independent of background carbonate chemistry. Our study builds on the foundational work of Feely et al.[26] and Fassbender et al.[24] that described key processes, and extends this analysis through time, permitting us to disentangle process-specific signals obscured in short-term records and resolve their impact on carbonate chemistry controls in the CCS over the 20th century.

## Results and discussion
### A century of acidification in the Salish Sea
Focusing first on the primary geochemical record in the Salish Sea, we find that the mean $\delta^{11}$B of modern corals is significantly lower than the

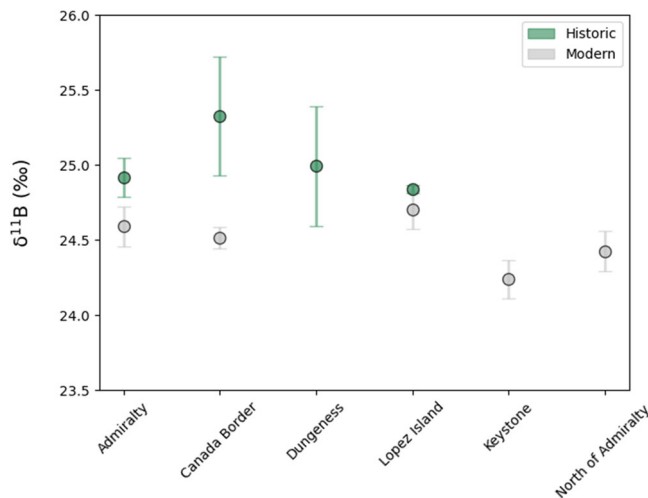

**Fig. 2 | Location-averaged coral skeletal δ11B measurements from the Salish Sea.** Historic (green) skeletal δ11B data from the Salish Sea are significantly higher than modern (gray) samples from the same locations. Sample sizes: Salish Sea historic, $n = 13$; Salish Sea modern, $n = 52$. Error bars represent the standard error of the mean for each location. For sites where only one coral was sampled, the population standard deviation was applied. This value was calculated as the average of the sample standard deviation across all locations with 8 or more corals. Sample locations are shown in Supplementary Fig. 1, and individual coral data points are detailed in Supplementary Fig. 2.

mean of historic samples from the same locations ($p = <0.001$; Fig. 2). We translate the measured decrease in coral δ11B into implied changes in carbonate chemistry over the last 130 years using a high-precision δ11B calibration from lab-cultured corals of the same species[27] ("Methods" section). Because our culture calibration and analogous research in tropical corals[28] show that coral δ11B is sensitive to both pH and dissolved inorganic carbon (DIC), we represent the way that δ11B constrains seawater carbonate chemistry as a line in pH (total scale) vs. DIC space (Supplementary Fig. 4, "Methods" section Eqs. 1 and 2). The pH-DIC lines described by our coral data in Supplementary Fig. 4 are nearly parallel with the contour lines corresponding to specific seawater $pCO_2$ values, which means we can make robust estimates of changes in $pCO_2$ over the past century (hereafter denoted Δ: $\Delta X = X_{modern} - X_{historic}$) in the Salish Sea regardless of the process causing the shift in ocean chemistry or changes in DIC ("Methods" section). The coral record indicates that $pCO_2$ increased by $172 \pm 41$ μatm between the 1890s and 2020, exceeding the rise in atmospheric $CO_2$ of 120 ppm over the same time period (1900–2020). The implied pH change over the industrial era is −0.095 (standard error of the difference between means $\sigma_{M_{Historic} - M_{Modern}} = 0.025$). Importantly, had acidification in the Salish Sea followed the pace of atmospheric $CO_2$ (our baseline case), ΔpH would be only −0.06 ("Methods" section). The $pCO_2$ and pH signals from the coral records indicate that the Salish Sea experienced amplified acidification over the 20th century. Even though the Salish Sea has naturally low pH and high $pCO_2$ and is a net source of $CO_2$ to the atmosphere, our coral records imply that these highly acidified waters not only became even more acidified over the last 130 years, but exceeded the magnitude of acidification implied by atmospheric $CO_2$.

To further interpret our record in terms of absolute pH, we align the modern corals with nearby modern observational data, which are offset by −0.1 ("Methods" section). We attribute this difference to culture conditions described in Gagnon et al.[27] to establish the δ11B calibration, which do not exactly match the natural wild coral setting (Supplementary Methods 1.3). This does not affect the interpretation of data from the Salish Sea, where we have historic-modern pairs, but applying this same offset to all coral samples anchors the historical

West Coast coral data where modern corals have not been collected, so they can be compared to modeled historic values and other proxies.

While different oceanographic processes influence the Salish Sea and CCS, the Salish Sea represents an extension of the CCS because the Salish Sea deep waters are sourced from upwelled water masses of the northern CCS. Therefore, our coral record serves as an indirect indicator of general shifts in seawater chemistry of the CCS. As an additional, independent test for amplified acidification across the CCS and in the Salish Sea, we complement our coral data with an evaluation of processes that influence estuarine acidification (Supplementary Discussion 1.1) and a modeling approach. Using an eddy-resolving physical-biogeochemical ROMS that extends along the CCS, we conducted two hindcast simulations spanning 1891–1904 and 1994–2007, hereafter referred to as historic and modern simulations, respectively ("Methods" section). These ROMS model datasets were compared to corals from the West Coast (discussed below) and were also used to set the composition of upwelled water entering a box model of the Salish Sea, which is not resolved in the ROMS model (Supplementary Discussion 1.2). To account for the temporal offset between the modern model (1994–2007) and modern corals (2020), during which additional atmospheric carbon accumulation occurred, we augment modeled $pCO_2$ from 2007 to 2020 to match the collection date of the modern corals (Supplementary Discussion 1.2). The Salish Sea box model approach predicts that seawater $pCO_2$ increased by $185 \pm 15$ μatm between the 1890s and 2020, agreeing with the magnitude of $\Delta pCO_2$ reconstructed using corals. Both the box model and our Salish Sea corals reveal a larger $\Delta pCO_2$ than that implied by atmospheric $\Delta CO_2$ and support amplified acidification in this region.

## Amplified acidification in the California Current System
We then expanded our analysis of historic acidification to the broader CCS using the same hindcast ROMS simulations. While the complex physics captured by the model results in spatial heterogeneity as in the real ocean, robust patterns in the depth structure of acidification are evident ("Methods" section, Supplementary Fig. 7). The well-ventilated upper 25 m of the nearshore CCS (<100 km) show 20th-century changes in the carbonate system that closely track the atmospheric increase in $CO_2$. Below this surface mixed layer, $pCO_2$ changes outpace the atmosphere, indicative of amplified acidification. Averaging the CCS across 50–200 m depth, the ocean has outpaced the atmospheric $pCO_2$ rise by 50%. The magnitude of $\Delta pCO_2$ increases with depth below 25 m, reaching a maximum at 100–125 m (Supplementary Fig. 8).

To assess the accuracy of the historic model, we introduce a record of $pCO_2$ based on skeletal δ11B of *B. elegans* along the US West Coast ("Methods" section, Supplementary Tables 3, 4). Our coral record presents general patterns and trends of acidification that are reproduced in the model (Fig. 3). The corals collected in the upper 25 m suggest that $pCO_2$ has followed the atmospheric rise in $CO_2$ since the 1890s, aligning with modeled predictions in the upper ocean. Below 50 m, every coral record suggests that the magnitude of 20th-century acidification has matched or exceeded the rise in atmospheric $CO_2$. The corals demonstrate that the CCS has indeed experienced amplified acidification over the industrial era, and even capture the trend of stronger amplification of acidification with depth.

In addition to these overall trends, it is worth noting specific differences between coral records and model simulations. For instance, the corals at several locations collected between 100 and 125 m outpace modeled $\Delta pCO_2$, suggesting even more amplified acidification. Our coral data from the Santa Barbara Basin (SBB) deviate most from the model and indicate an even lower historic $pCO_2$ than that represented in the model, and thus a larger relative increase in $pCO_2$ (Fig. 3; $n = 5$). Notably, one of the only previously published records of SBB carbonate chemistry over the industrial era also indicates lower historic $pCO_2$, like our historic corals[29] (Supplementary Discussion 1.4).

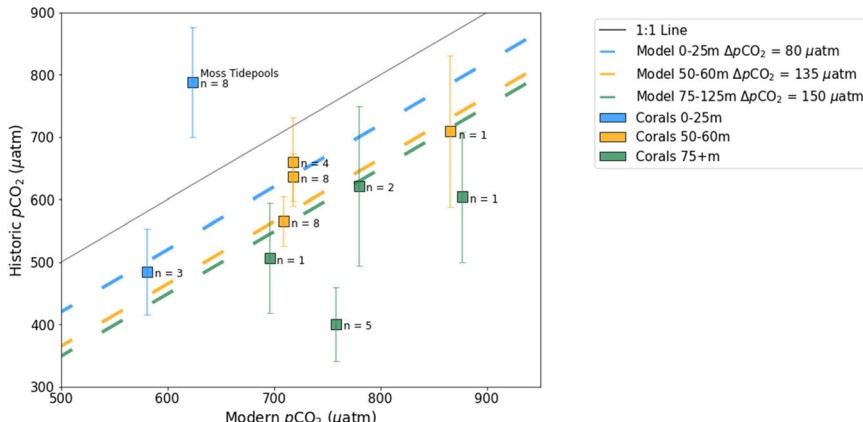

**Fig. 3 | Geochemical and model records of 20th-century acidification in the California Current.** Comparison between modeled historic $p$CO$_2$ and measured historic coral $p$CO$_2$ in the CCS as a function of modern modeled $p$CO$_2$. Lines: The solid 1:1 line indicates no acidification over the past century (historic $p$CO$_2$ = modern $p$CO$_2$). The magnitude of boreal summertime acidification over the 20th century ($\Delta p$CO$_2$) produced in the model is dependent on depth, represented by the blue, orange, and green dashed lines to reflect different depth bins, which is consistent with marine biogeochemical processes. Acidification in the upper 25 m (blue line) has largely followed the trend of atmospheric CO$_2$ ($\Delta p$CO$_2$ = 80 µatm) over the 20th century (1900–2000). Modeled $\Delta p$CO$_2$ below 50 m outpaces the rise in atmospheric CO$_2$ and suggests that the California Current has experienced amplified acidification. As depth increases in the CCS, the model implies that the strength of amplified acidification also increases. Coral data: Squares represent the average coral data from individual locations and align with general patterns of modeled acidification ("Methods" section, Supplementary Data 1). The depths of coral collection are color-coded to match the depth of modeled data. Vertical error bars represent standard errors of the mean. Error bars for locations with only one coral represent the population standard deviation, inferred by locations with 8+ samples ("Methods" section). $n$ = number of corals sampled at each location. The width of the rectangles represents the average modern model standard error of the mean across locations ($x$ = 6 µatm). Only one coral location implies a lower modern $p$CO$_2$ than historic $p$CO$_2$ (USNM 78638); these corals were collected from a tide-pool near Moss Beach, California, where the high variability associated with the intertidal zone likely influenced the $\delta^{11}$B skeletal signatures.

Although our data from this region are too sparse to be conclusive, it is possible that regional processes that amplify acidification in the SBB even more than the rest of the CCS are not captured in the model and warrant further investigation.

Overall, the coral record reproduces broad patterns of acidification in the biogeochemical model, and they collectively suggest that the CCS has experienced amplified acidification over the 20th century. To probe potential drivers of this acidification, we test for shifts in upwelling and biological activity over the past century using the ROMS simulations ("Methods" section). A climatology of modeled upwelling velocity suggests that upwelling has weakened slightly during the summer months and its onset has shifted earlier in the year over the 20th century. Comparing 7-year averages of upwelling velocity between the historic and modern eras, the decline in the CCS is statistically significant but only represents a 2% change in strength and is unlikely to have driven the observed amplification of acidification ("Methods" section). Changes in the biological pump could also affect the integrated impact of remineralization on DIC. We quantified this change in integrated remineralization using apparent oxygen utilization. Over the 20th century, changes in remineralization can account for 15% of modeled ΔDIC in the CCS on average, but cannot explain the driving force of amplified acidification (Supplementary Fig. 9).

Much of the amplified increase in $p$CO$_2$ that we observe in the CCS can be explained by thermodynamic buffering effects. The isopycnals corresponding to upwelled water masses outcrop at open ocean locations, distant from the CCS. Here, we assume that water masses interact with the overlying atmosphere such that initial $p$CO$_2$ follows increasing atmospheric $p$CO$_2$. After subduction, remineralization increases DIC, $p$CO$_2$, and acidification as the water circulates along an isopycnal and is eventually upwelled in the CCS. Increasing initial preformed DIC in response to rising atmospheric $p$CO$_2$ has an outsized effect on $p$CO$_2$ and pH in upwelled water due to a reduction in buffering capacity, even if remineralization and circulation rates remain the same (Fig. 1c). This amplification highlights the need to assess rates of acidification with careful consideration of the mean state of the carbonate system, especially in extreme domains in carbonate chemistry space, such as high-carbon upwelling systems. While local physical and biogeochemical factors in other upwelling systems may exert additional influences, the chemical principles underlying the amplification are inherent to all upwelling regimes, underscoring the global implications of this chemical non-linearity.

Averaging across the CCS, estimates of ΔDIC derived from equilibration of atmospheric CO$_2$ at outcropping regions and accompanying thermodynamic buffering effects ("Methods" section Eq. 4) can explain the modeled ΔDIC to within ±15% relative error ("Methods" section, Supplementary Fig. 10). This analysis shows that atmospheric CO$_2$ and thermodynamic buffering effects drive the magnitude of acidification in large areas of economic and ecological value. The remaining residual leaves room for second-order effects such as shifts in upwelling strength and remineralization rates, as described above. Notably, this analysis also underscores the importance of water mass age when attributing biogeochemical change to anthropogenic carbon ("Methods" section). Given the rapid increase in atmospheric CO$_2$ over the past 50 years, considering water mass age in upwelling regions markedly changes the accumulation of anthropogenic carbon, as deeper, older water was in contact with its contemporary atmosphere less recently[30]. Acidification in the CCS lags the imprint of modern atmospheric CO$_2$ due to water mass age, suggesting that acidification will continue to amplify and worsen in the coming decades.

## Predicting future OA in the California Current

Our historic coral data build confidence in the patterns and magnitude of acidification produced by our model, including amplified acidification. From this foundation, we analyze projections for future acidification in the CCS under the representative concentration pathway 8.5 (RCP 8.5). Although RCP 8.5 represents a high-forcing scenario, there is value in bracketing the upper end of potential future states[31]. These analyses build on a multi-resolution model study that assessed the value of resolving coastal processes in predicting future change[15]. Amplified acidification is projected to continue in the subsurface CCS throughout the 21st century (Fig. 4d, e). Ominously, $\Delta p$CO$_2$ in the CCS is estimated to continue to outpace the atmospheric rise in CO$_2$ in the

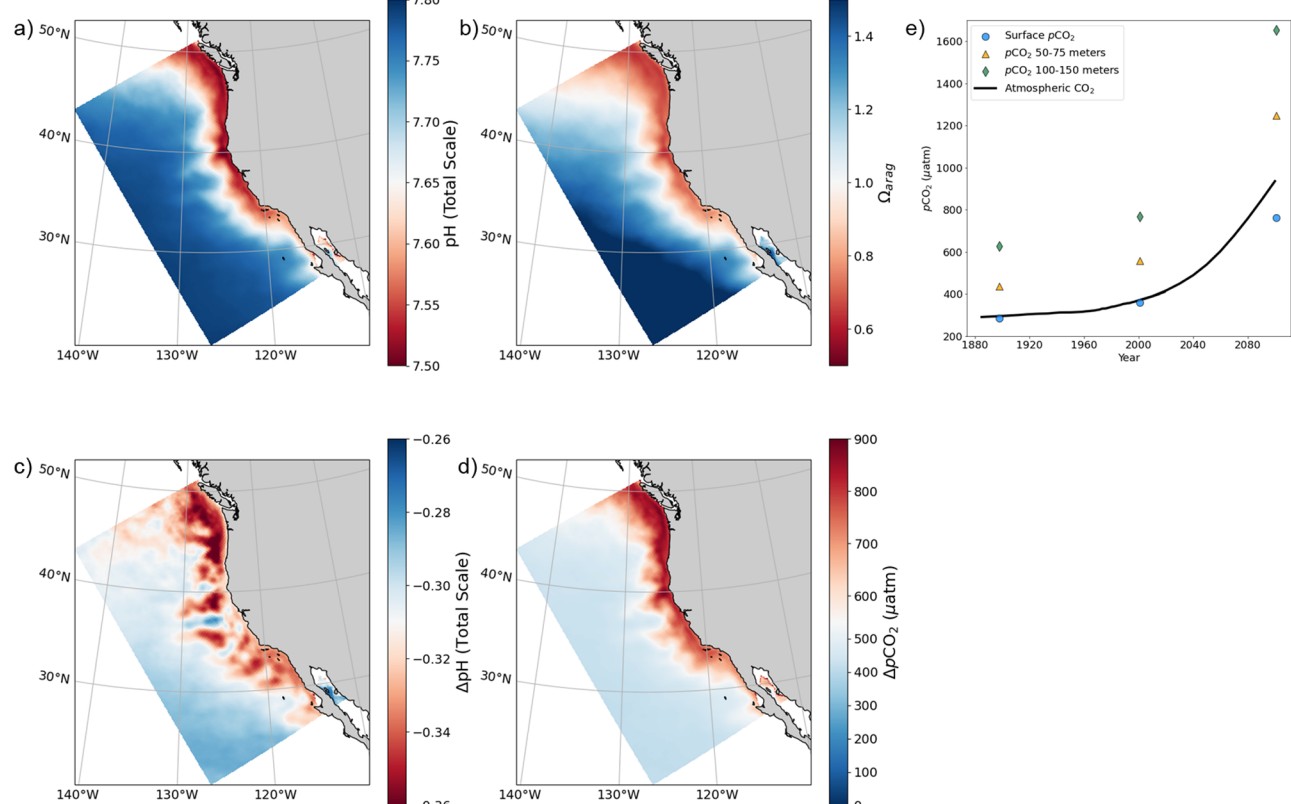

**Fig. 4 | Future acidification in the CCS following the RCP 8.5 emissions scenario.** 7-year averages at 75 m for 2097–2104 of **a** pH (total scale) and **b** aragonite saturation state ($\Omega$). 21st-century changes in **c** pH and **d** $pCO_2$. The color bar in (**d**) is centered at 550 µatm to reflect the expected rise in atmospheric $CO_2$ following RCP 8.5 over the 21st century; red colors indicate amplified acidification. **e** 7-year modeled averages of the CCS seawater $pCO_2$ at the surface (blue), 50–75 m (orange), and 100–150 m (green) in the historic (1897–1904), modern (2000–2007), and future (2097–2104) simulations. The black line represents atmospheric $CO_2$ following RCP 8.5. The subsurface CCS is expected to experience amplified acidification relative to the atmospheric rise in $CO_2$.

coming century by 20% between 50 and 75 m and 60% between 100 and 150 m. pH is expected to decline by –0.30 below 50 m, equivalent to the predicted change for the global surface ocean[32]. While comparable in magnitude to $\Delta$pH, the increase in hydrogen ion concentration [H+] in deeper waters is nearly double that at the surface because of the logarithmic nature of the pH scale[33]. The CCS is projected to experience low aragonite saturation states ($\Omega$ < 1) and absolute pH levels as low as 7.5 at 75 m due to the already highly acidified conditions of the CCS. Local marine organisms of economic, cultural, and biological importance are predicted to experience significant physiological challenges in response to the increasingly acidified conditions[6]. For instance, Dungeness crab, the largest US West Coast fishery by revenue, is expected to experience reduced growth and survival in early life stages as acidification progresses[6,34]. While the surface ocean across the model domain largely followed atmospheric $CO_2$ over the 20th century, surface $\Delta pCO_2$ lags the atmospheric increase in the 21st century, indicative of the surface ocean's limit in mitigating the accumulation of atmospheric $CO_2$ (Fig. 4e). Acidification of this magnitude is predicted to have profound impacts on ecosystems of the CCS in the 21st century.

## Methods

### Sample measurements: sample collection
Historic samples of *Balanophyllia elegans* (*B. elegans*) were collected live by the R/V *Albatross* between 1888 and 1894, and by Ernest Harrison Quayle in 1932. All historic samples were archived in the Smithsonian National Museum of Natural History (Supplementary Fig. 1, Supplementary Data 1). Samples were dried upon collection and archived dry following Smithsonian preservation guidelines. In the

Salish Sea, a total of $n = 13$ historic corals were collected from five sites. A total of $n = 41$ historic corals were collected along the outer US West Coast from 10 sites that span from Vancouver Island, British Columbia, to the Channel Islands, California. These corals were collected in the 1890s from sites that would coincidentally become the major federal marine sanctuaries of the US West Coast.

Modern corals ($n = 42$) were collected from the same locations as the historic corals in the Salish Sea aboard the University of Washington R/V *Rachel Carson* in Fall 2020 (Supplementary Fig. 1, Supplementary Data 1). These corals therefore reflect conditions from approximately 2010 to 2020.

While the collection dates of most coral locations are known, and all corals were living, the age of each coral is unknown. The age of the historic coral at the time of collection is unlikely to affect the study outcome, as pH values were assumed to be relatively stable. Additionally, all of the historic corals in the CCS were collected within 1.5 years of each other (September 1888–March 1890). The age of the modern corals is also unlikely to impact the modern pH values, as they represent conditions over a narrow time window (5–10 years). Any uncertainty in the exact age, and therefore presumed pace of acidification over this period, is within the measurement error of the corals.

### Sample measurements: boron isotope analysis
Skeletal sections of ~5 mg $CaCO_3$ were sampled along the primary growth axis from each coral using a Dremel tool approximately ~0.5–1 cm deep. The sampling scale is large relative to skeletal microstructures like centers of calcification, which likely means that their effects are averaged out and homogenized within the bulk sample. This is supported by the systematic trends and tight precision of even

smaller samples from Gagnon et al.[27]. Additionally, the Gagnon et al.[27] calibration suggests that intraspecies variability is small compared to environmental effects. The collected material from each coral represents an average over the entire growth period of the coral. It has been estimated that the average life span for *B. elegans* in the California Current is 6.5–11 years[35]. This time window is sufficiently narrow that any secular change in pH due to acidification within this period is within the measurement error of the corals. By sampling along the whole primary growth axis, we assume that different coral skeletal components are evenly represented in each sample.

The skeletal subsamples were then analyzed for their boron isotope composition ($\delta^{11}$B) in the St Andrews Isotope Geochemistry laboratories (STAiG) of the University of St Andrews, UK. Samples were first rinsed with ultra-pure water (Milli-Q, 18.2 Ω) and subsequently oxidized to remove any organic fractions using 1% $H_2O_2$ buffered with 0.1 M $NH_4OH$ at 80 °C following routine sample preparation protocols for boron isotope analyses (e.g., Rae[36]). Historic samples were dried upon collection and archived dry following Smithsonian preservation guidelines. All samples of the same weight (~5 mg) were treated identically during cleaning to minimize the risk of preferential dissolution of specific coral skeletal components biasing results. Clean samples were dissolved in distilled 0.5 M $HNO_3$ with the aid of ultrasonication. A small aliquot of the dissolved sample (<10% volume) was measured on an Agilent 8900 Triple Quadrupole ICP-MS to determine the boron content in the sample along with major, minor, and trace element composition. The rest of the dissolved sample was separated from the carbonate matrix on an Amberlite IRA 743 boron-specific anionic exchange resin[37,38], following the batch protocol of Trudgill et al. 2024[39]. Purified boron solutions were spiked with HF to aid boron washout and measured in a 0.5 M $HNO3$ + 0.3 M HF matrix[40,41] on a Thermo Scientific Neptune Plus MC-ICP-MS equipped with an HF-resistant sample introduction kit and $10^{13}$ Ω resistors (see Crumpton-Banks et al.[42] for further details). Instrumental mass bias was corrected by standard sample-bracketing with a 15 ppb NIST SRM 951 solution. Total procedural blanks contained <20 pg B (*n* = 4), and hence were negligible against the typical sample size of ~20 ng B (<0.1%). Full procedural uncertainty was assessed using repeat measurement of NIST RM 8301 (Coral) consistency standard run during this analytical campaign, yielding an average value of 24.28 ± 0.12‰ (2 SD, *n* = 8) that is consistent with long-term lab reproducibility and interlaboratory reference values[43] (24.17). Skeletal data are summarized in Supplementary Data 1.

## Data processing: the boron isotope-pH proxy

Paleoproxies provide an invaluable tool to reconstruct historic ocean conditions and contextualize current climate change. Stable boron isotope ratios ($\delta^{11}$B) are recorded in the skeletons of marine calcifiers such as corals, which function as time capsules of past carbonate system chemistry[44]. We utilized this method to reconstruct the historic state of the CCS through measurements of *Balanophyllia elegans*, a common cold-water orange cup coral that spans a wide geographic range from Southern Alaska to Baja California.

To test the null hypothesis that $\delta^{11}$B of borate ion has not changed over the past century in the Salish Sea, we conduct a two-sample *t*-test to determine any difference between historic and modern $\delta^{11}$B. Averaging data across sample locations in the Salish Sea (Supplementary Methods 1.1), there is a significantly higher $\delta^{11}$B in the historic corals (M = 24.94, SE = 0.10) compared to $\delta^{11}$B measurements from modern corals (M = 24.52; SE = 0.06); *t*(63) = 3.53, *p* = <0.001. Assuming that the physiology of coral calcification and total seawater $\delta^{11}$B have not changed over the past century, we can reject the null hypothesis and conclude that $\delta^{11}$B of borate ion incorporated in the coral skeletons has declined as a result of declining seawater chemistry (Supplementary Fig. 2).

## Data processing: coral-based estimates of $p$CO$_2$ in the Salish Sea

We then translate the coral data into constraints on past seawater carbonate chemistry to add meaning to this measured $\Delta(\delta^{11}$B) ($\Delta$ = modern – historic). In a previous study, the same species of corals (*B. elegans*) was cultured under controlled conditions to produce a calibration curve that links skeletal $\delta^{11}$B and seawater pH, the predominant metric to assess ocean acidification[27] (Eq. 2). The same culture study also uncovered a subtle but significant relationship between skeletal $\delta^{11}$B and dissolved inorganic carbon (DIC), even at constant pH[27] (Eq. 1). A similar relationship between $\delta^{11}$B, pH, and DIC is apparent in culture studies of tropical corals[28].

$$\delta^{11}B_{correction} = (DIC - 2.5\,mmol/kg) \times 1.08 \qquad (1)$$

$$pH_{TOTAL} = \frac{\delta^{11}B_{measured} + 1.08 \times DIC + 7.2}{4.33} \qquad (2)$$

Albeit minor relative to the skeletal $\delta^{11}$B–pH relationship, the impact of DIC on skeletal $\delta^{11}$B is nonetheless considered in our interpretation of the coral data (Eq. 2). Because two environmental parameters exert control on $\delta^{11}$B, we explore the coral record in pH vs DIC space as shown in Supplementary Fig. 4. Historic and modern skeletal $\delta^{11}$B measurement averages are indicated by lines in this space. Fortuitously, we find that $\delta^{11}$B provides a tight constraint on $p$CO$_2$ regardless of changes in DIC, as can be seen on these plots. Over a large range of DIC values, lines of constant $\delta^{11}$B are nearly parallel with the contour lines corresponding to specific seawater $p$CO$_2$ values (Supplementary Fig. 4). While individual contours are parallel between $p$CO$_2$ and $\delta^{11}$B over a wide range of DIC values, the spacing between contours does change subtly with absolute DIC. However, the effect of DIC falls within the measurement error and was not found to have a meaningful effect. Once a rough DIC is established for the modern ocean at coral collection locations from observational (bottle) data, the $\Delta p$CO$_2$ between historic and modern eras is simply the distance in space between the modern and historic constant $\delta^{11}$B lines. Note that DIC does not need to be known well, as it is just used to generally indicate the region of carbonate system space for making the plots used here.

We use climatological means of bottle data from the Washington Ocean Acidification Center cruises to obtain modern values of mean DIC and pH at the sites of coral collection[45] (Supplementary Methods 1.2). Most of our Salish Sea coral samples were collected near Admiralty Inlet, a narrow channel at the eastern terminus of the Strait of Juan de Fuca through which most of the seawater that fills Puget Sound flows. Tidal mixing over a shallow glacial sill leads to similar seawater properties through much of the water column at Admiralty Inlet, meaning the properties are generally homogeneous with depth. Climatologies of modern carbonate chemistry parameters for Admiralty Inlet at the coral collection depth of 70 m were generated from several years of bottle data. There is a seasonal pattern of low TA and DIC during the winter months, followed by high TA and DIC during the fall (Supplementary Methods 1.2; Supplementary Fig. 13). We conducted a sensitivity analysis using both annual and summertime (JJA) averages for modern DIC and found only a minor difference in our reconstructed $\Delta p$CO$_2$ results for the Salish Sea. This is in part because summer and annual climatological means of pH averages are approximately equivalent in the Salish Sea. Armed with this constraint on typical DIC, we can make robust estimates of $\Delta p$CO$_2$ in the Salish Sea regardless of the process that drives the 20th-century change ($\Delta p$CO$_2$ = 172 µatm; standard error of the difference between means $\sigma_{M_{Modern} - M_{Historic}}$ = 41 µatm).

We assessed the temperature effect on the *k*-value by conducting a sensitivity analysis on our $p$CO$_2$ calculations. Temperature values were based on summertime averages from bottle data collected at the

Washington Ocean Acidification Center (WOAC) station 21 (Supplementary Table 1), located near a majority of coral sampling sites. Under the assumption that temperature remained constant over the past century, we calculated $\Delta pCO_2 = 172$ µatm. However, if we assume that historical temperatures were 1 °C cooler, then $\Delta pCO_2$ increases slightly to 176 µatm. This small temperature effect is well within the uncertainty range of the coral records (±41 µatm). Furthermore, if temperatures have indeed warmed over the past century, our acidification estimates would be even larger and more extreme. We adopt a conservative approach and assume a constant temperature over the past century, and thus report $\Delta pCO_2 = 172$ µatm in the Salish Sea.

### Data processing: coral-based estimates of pH in the Salish Sea

The modern observational data provides additional information that we can use to better calibrate our proxy and to estimate $\Delta pH$ for the Salish Sea. Comparing modern observational DIC and pH data (red circle, Supplementary Fig. 4, Supplementary Methods 1.2) to modern $\delta^{11}B$ (gray line, Supplementary Fig. 4), we found that a pH offset of −0.10 was necessary to align the two datasets. We attribute this difference to culture conditions not exactly matching the natural wild coral setting. To calibrate the coral data, we apply this pH offset of −0.10 to all our coral data, including our historic and modern records from the Salish Sea and historic records from the US West Coast. This offset does not impact the interpretation of data from the Salish Sea, where we have historic-modern pairs, but applying this same offset to all coral samples anchors the historical West Coast coral data so they can be accurately compared to modeled historic values and other proxies. We also find that the summer and annual climatological pH averages are approximately equivalent in the Salish Sea, so applying an offset informed by the observational summertime average instead of choosing the annual average has little impact. Nonetheless, we operationally had to make a choice between summertime and annual averages. We used summertime for our offset correction because there is evidence that skeletal growth of *B. elegans* and other closely related cold-water coral species is biased towards summer[35,46,47] (Supplementary Methods 1.3).

Unlike $\Delta pCO_2$, to most accurately reconstruct $\Delta pH$, the process driving this shift in pH should be known. Different biogeochemical processes lead to different paths in pH-DIC space between the $\delta^{11}B$ curves representing modern and historic coral data. This choice of path will result in slightly different implied $\Delta pH$ values (but does not significantly impact $\Delta pCO_2$). For example, if most of the Salish Sea seawater changes captured by $\delta^{11}B$ were driven by accumulation of anthropogenic $CO_2$, then the implied pH would be slightly smaller than if seawater chemistry changes were primarily driven by shifts in ecosystem calcification and $CaCO_3$ dissolution (see colored paths in Supplementary Fig. 4). We assume that $CO_2$ invasion is the main driver of the 20th-century change (as explained in Supplementary Discussion 1.1), which constrains the historic estimate of DIC and allows us to calculate an absolute estimate of historic pH. This biogeochemically reasonable assumption also leads to a smaller, more conservative $\Delta pH$ estimate than other processes, like shifts in calcification and $CaCO_3$ dissolution. Note again that the sensitivity of $\Delta pH$ to the choice of processes is small compared to other sources of uncertainty. The intersections of the $CO_2$ invasion line and historic and modern coral records in Supplementary Fig. 4 imply $\Delta pH = -0.095$ (standard error of the difference between means $\sigma_{M_{Modern}-M_{Historic}} = 0.025$) and $\Delta DIC = 28$ µmol/kg. This estimate of $\Delta DIC$ is comparable to estimates of industrial era $\Delta DIC$ in the Salish Sea from a high-resolution biophysical modeling study[48].

Despite the coral-based $\Delta pH$ and $\Delta pH$ of the open surface ocean over the past century being comparable in magnitude (~−0.1), the $\Delta pH$ reported in the Salish Sea subsurface still supports amplified acidification due to the logarithmic nature of the pH scale[33]. The mean chemical state of the open ocean surface is vastly different than that at depth in the Salish Sea; therefore, the *same* amount of atmospheric $CO_2$ change will lead to *different* changes in pH dependent on the background carbonate chemistry of the system. The Salish Sea has a higher DIC:TA value than the open ocean surface and thus a lower buffering capacity. To understand the effect of this background state on subsequent changes in carbonate chemistry, we can calculate the predicted $\Delta pH$ in our 'baseline' scenario, defined as the magnitude of acidification associated with the uptake of atmospheric $CO_2$ over the industrial era ($\Delta CO_2$ atm = 120 ppm between 1900 and 2020). Feely et al. 2010[49] Table 1 provides historic and modern estimates for DIC, TA, and pH at depth at Admiralty Inlet, where the majority of our corals were collected. From these variables, we can calculate historic $pCO_2$ and estimate modern pH assuming modern $pCO_2$ = historic $pCO_2$ + 120 µatm (the change in atmospheric $CO_2$ over the 20th century). The predicted $\Delta pH$ for this baseline scenario is −0.06, far smaller than the −0.095 change suggested by the coral record. Furthermore, the absolute change in [H+] is 1.5 times larger in the Salish Sea (associated with the baseline scenario yielding a −0.06 pH decline) versus that in the open ocean (associated with −0.1 pH decline), given the logarithmic nature of the pH scale. For these reasons, the $\Delta pH$ in the Salish Sea is in fact amplified when compared to the magnitude of acidification that follows the pace of atmospheric change.

### Data processing: coral-based estimates of acidification in the CCS

West Coast $\delta^{11}B$ data are interpreted in a similar way as our Salish Sea corals, including the application of a −0.10 pH offset. While the summer and annual pH averages are approximately equivalent in the Salish Sea, locations along the West Coast have different climatologies. The West Coast locations experience low pH earlier in the season compared to the Salish Sea (Supplementary Fig. 3). As a result, the annual and summer pH averages along the West Coast are different, whereas they are approximately equivalent in the Salish Sea. Given that different choices lead to smaller or larger offsets between the West Coast model and coral data, different assumptions can affect the extent of overlap between the coral data and the model. However, all but one combination of these various assumptions is consistent with amplified acidification. Furthermore, the one outlier scenario results in historic $pCO_2$ values that are unrealistically high, so we feel confident in excluding this interpretation from our analysis.

For the reasons described previously, the $pCO_2$ of historic corals in the CCS is well constrained by historic $\delta^{11}B$ measurements. These reconstructed $pCO_2$ values are independent of our model and can be compared to modern data, modeled or observational, as well as other proxies. The results of our study are primarily based on these independent $pCO_2$ reconstructions.

It is also convenient to estimate past pH from historic $\delta^{11}B$ measurements, even if this requires some other constraint on past DIC. We use an average DIC extracted from the ROMS model by calculating a 7-year summertime average of historic DIC in the nearshore CCS over 0–125 m depth (last 7 years of the historic run). The average DIC value is substituted into Eq. 1, the $\delta^{11}B$-DIC correction equation described above. The $\delta^{11}B$ correction values and location-averaged $\delta^{11}B$ measurements are then substituted into Eq. 2 to obtain DIC-corrected pH values at each coral location.

Standard errors of the mean were calculated for pH and $pCO_2$ at every location. For locations with only one coral, the standard error of the mean is equal to the population standard deviation, inferred by calculating the average standard deviation at locations with 8+ samples. The $pCO_2$ data and standard error of the mean values described here represent the y-axis values in Fig. 3. The coral-based calculations of historic pH and $pCO_2$ can be found in Supplementary Table 3.

## Data processing: physical-biogeochemical ROMS of the CCS

A regional eddy-resolving physical-biogeochemical Regional Ocean Modeling System (ROMS) was used to assess the change in carbonate chemistry along the US West Coast over the 20th century. Two hindcast simulations were analyzed, spanning 1994–2007 in the modern era and 14 years representing the end of 19th-century conditions as in the historic era. The horizontal grid is $322 \times 450$ cells with a resolution of $dx = 12$ km, $dy = 12$ km, $dt = 32$ min, and $dz = 42$ terrain-following sigma layers with decreasing layer thickness towards the surface. For the purposes of our analyses, the model results are interpolated to 25-m depth increments and saved as monthly averages. While the modern model has been evaluated against observational data and faithfully reproduces the modern CCS mean state (Supplementary Fig. 6), no long-term dataset exists across the CCS of carbonate chemistry parameters in the early 1900s to assess the accuracy of the historical run. Additional details on the model configuration and reproducibility can be found in Renault et al.[50] and Deutsch et al.[51]. The input for the historical run is based on the input for the hindcast run, with long-term trends of temperature and DIC between the two periods removed. The interannual variability of the surface forcing is the same in both runs. Long-term trends in radiation and relative humidity were removed for the historical run.

The model domain was divided into nearshore (<100 km) and offshore (>200 km) regions during analysis to capture variability in biogeochemical patterns between the coastal CCS and the open ocean. All averages of modeled parameters represent 7-year summertime or annual averages in the historic (last 7 years of the historic run) and modern (December 2000–November 2007) simulations. Only the final seven years of the simulations were included in the calculations to remove model biases due to spin-up and deviations from the mean state, such as the 1997–1998 El Niño.

To compare West Coast coral and model data, the coral data are compared to a modeled summer average, since the corals grow with a seasonal bias in the summer (Supplementary Methods 1.3).

## Data processing: model estimates of acidification in the CCS

Model predictions for 20th-century changes in pH, DIC, and $p\mathrm{CO_2}$ are assessed along the US West Coast (Supplementary Fig. 7). We primarily analyze acidification in terms of $p\mathrm{CO_2}$ as opposed to pH, allowing us to directly compare magnitudes of acidification independent of background carbonate chemistry. Model estimates of $\Delta p\mathrm{CO_2}$ are also obtained from several depth regimes in the nearshore CCS to align with the coral depths of collection. Above 25 m depth, $\Delta p\mathrm{CO_2}$ from 1890 to 2005 slightly lags the atmospheric rise in $\mathrm{CO_2}$ ($\Delta p\mathrm{CO_2} = 75$ µatm; $\Delta p\mathrm{CO_{2\,atm}} = 80$ µatm). At 50–60 m, $\Delta p\mathrm{CO_2}$ in the nearshore CCS is 120 µatm; at 75–125 m depth, $\Delta p\mathrm{CO_2}$ in the nearshore CCS is 140 µatm. Summertime averages of $\Delta p\mathrm{CO_2}$ are also evaluated to compare the coral records, which grow with a seasonal bias (Supplementary Methods 1.3). In the upper 25 m of the water column, $\Delta p\mathrm{CO_2}$ from 1890 to 2005 in the summer months follows the atmospheric rise in $\mathrm{CO_2}$ ($\Delta p\mathrm{CO_2} = 80$ µatm). At 50–60 m depth during the summer, $\Delta p\mathrm{CO_2}$ in the nearshore CCS is 135 µatm. At 75–125 m depth, $\Delta p\mathrm{CO_2}$ in the nearshore CCS is 150 µatm. The above summertime $\Delta p\mathrm{CO_2}$ values at 0–25 m, 50–60 m, and 75–125 m are reproduced in Fig. 3 to represent average model predictions of acidification in the nearshore CCS.

Since no historic-modern pairs exist along the outer West Coast, modeled historic $p\mathrm{CO_2}$ and measured historic coral $p\mathrm{CO_2}$ are both compared to modern modeled $p\mathrm{CO_2}$. Both historic and modern model $p\mathrm{CO_2}$ averages are calculated over a 7-year period at every coral location from December 1897 to November 1904 and December 2000 to November 2007, respectively. Since the corals are presumed to grow with a seasonal bias during the summer due to optimized feeding and nutrient supply (Supplementary Methods 1.3), the modern model averages include data only from summer. In all but one location, the season of environmental conditions that aligns with optimal growth

corresponds with June, July, and August. Our northernmost location (USNM 92621), however, shows a different climatology where optimized nutrient conditions are delayed. This is likely due to the cumulative seasonal respiration, which contributes to acidified and hypoxic conditions in the northern CCS through the upwelling season, where there is longer retention[25,34,52]. For this location, we select a 3-month window including July, August, and September. The $p\mathrm{CO_2}$ averages are calculated over a 3-cell × 3-cell area (36 km × 36 km) centered on the grid cell nearest the coral location.

The model output is interpolated to 25-m depth intervals. Even though most collection depths are within 5 m of a 25-m increment, USNM 92626, USNM 36416, and USNM 92625 are located halfway between depth bins (Supplementary Data 1). In these cases, the model is linearly interpolated between the two adjacent depth bins to account for $p\mathrm{CO_2}$ gradients with depth. The historic, coral-based $p\mathrm{CO_2}$ data are plotted against this modern, model-based $p\mathrm{CO_2}$ data in Fig. 3 to compare model and geochemical estimates of 20th-century acidification.

## Data interpretation: quantifying acidification in the CCS attributed to changes in upwelling and remineralization

To probe the dominant driver of 20th-century acidification, we test for shifts in upwelling and biological activity over the past century using our model. Upwelling strength was evaluated by comparing 7-year averages of upwelling velocity in the nearshore CCS (<100 km) at every depth bin in the historic and modern model simulations. A paired $t$-test reveals a statistically significant decline in upwelling velocity of 2% in the upper 200 m.

To capture the integrated impact of shifting upwelling patterns and remineralization rates on carbonate chemistry, we calculate the change in apparent oxygen utilization (AOU) over the 20th century.

To determine changes in remineralization, we calculate DIC attributed to respiration inferred by AOU and the stoichiometric Hedges ratio in both the historic and modern model simulations[53]. For AOU calculations, $\mathrm{O_2}$ saturation values are calculated at every grid cell in the nearshore CCS for both the historic and modern eras, informed by 7-year averages of temperature and salinity for the given grid cell[54]. Model $\mathrm{O_2}$ is subtracted from saturated $\mathrm{O_2}$ to give AOU, or $\mathrm{O_{2\,remineralization}}$, the oxygen consumed during the transformation of organic matter to inorganic carbon. To obtain DIC produced via remineralization, the AOU was multiplied by the Hedges[53] stoichiometric relationship $\mathrm{C/O_2} = 106/152.5$. DIC production attributed to respiration was compared in the historic and modern eras ($\Delta\mathrm{DIC_{remineralization}} = \mathrm{DIC_{remineralization\,modern}} - \mathrm{DIC_{remineralization,\,historic}}$) to determine 20th-century changes in remineralization. Supplementary Fig. 9 shows the fraction of $\Delta$DIC that can be explained by 20th-century changes in remineralization ($\Delta\mathrm{DIC_{remineralization}}/\Delta\mathrm{DIC}$), which is small relative to other sources of change. Thus, neither shifts in upwelling strength nor changes in remineralization can explain the dominant driver of 20th-century acidification in the CCS.

## Data interpretation: quantifying acidification in the CCS attributed to anthropogenic carbon

Anthropogenic carbon is transported to the CCS along isopycnals that initially outcrop in the open ocean. These water masses subduct at the outcrop regions where the water is in exchange with $\mathrm{CO_2}$ in the overlying atmosphere. To estimate the change in carbonate chemistry attributed to anthropogenic carbon that enters the ocean at outcrop regions, we calculate the $\Delta\mathrm{DIC_{preformed}}$ expected from just an increase in atmospheric $p\mathrm{CO_2}$:

$$\Delta\mathrm{DIC_{preformed}} = \mathrm{DIC_{preformed,\,modern}} - \mathrm{DIC_{preformed,\,historic}} \quad (3)$$

We assume for this simplified test that the increase in atmospheric $p\mathrm{CO_2}$ between the historic and modern eras translates into an equal

change in sea surface $p\mathrm{CO_2}$. This assumption does not necessarily imply equilibration, but it does make the simplifying assumption of a constant air-sea offset for the outcrop of a particular isopycnal with time. At each grid cell in the CCS, surface, preformed DIC at the outcropping region was calculated using (1) atmospheric $\mathrm{CO_2}$ and (2) a 7-year average of the grid cell TA. Note that the difference in preformed DIC between historic and modern is only weakly sensitive to the alkalinity of the preformed water, because contours of constant $p\mathrm{CO_2}$ are roughly parallel over the relevant range of carbonate system parameters. Thus, our results are robust to even large uncertainties in mean alkalinity or alkalinity changes.

Water mass age affects our calculations of preformed DIC because this age determines the date when the water mass was most recently in exchange with the atmospheric $\mathrm{CO_2}$, which has changed rapidly in the modern era. We assume a typical age structure for the CCS where deeper water masses are older. Our analysis was conducted assuming ages of 0, 13, and 25 years at 0–50 m, 75–125 m, and 150–200 m, respectively, to reflect varying time since last ventilation. In the modern model (2000–2007), these ages correspond with atmospheric $\mathrm{CO_2}$ concentrations of 376, 360, and 339 μatm for ~2005, 1995, and 1980, respectively (Switzerland Institute for Atmospheric and Climate Science, Annual $\mathrm{CO_2}$ Data). Given the rapid rise of atmospheric $\mathrm{CO_2}$ since the 1970s, accounting for water mass age is an important consideration when analyzing acidification patterns in the CCS. In the historic model (end of 19th century), atmospheric $\mathrm{CO_2}$ was relatively constant over the prior 25 years, so water mass age makes little difference in terms of anthropogenic carbon accumulation ($\mathrm{CO_{2\,atm}} = 296$ μatm).

The difference between modern and historic preformed DIC values represents the estimated ΔDIC attributed to anthropogenic carbon (ΔDIC$_{\mathrm{preformed}}$, Eq. 3). The ratio of estimated, preformed ΔDIC to modeled ΔDIC is calculated at every grid cell to determine how much modeled 20th-century changes can be explained by uptake of anthropogenic $\mathrm{CO_2}$. Supplementary Fig. 10 shows the percent that uptake of anthropogenic $\mathrm{CO_2}$ can explain modeled ΔDIC at 0–50 m, 75–125 m, and 150–200 m depth, assuming structured water mass age. Estimates of preformed ΔDIC derived from equilibration of atmospheric $\mathrm{CO_2}$ at outcropping regions and accompanying thermodynamic buffering effects can explain modeled ΔDIC within 15% relative error. The remaining residual leaves room for second-order effects such as changes in upwelling strength and biological activity, independently estimated based on AOU calculations and model-based upwelling parameters, described above.

Modeled and estimated acidification attributed to anthropogenic carbon were also compared in terms of $p\mathrm{CO_2}$ and pH. Estimates of modern DIC assuming atmospheric $\mathrm{CO_2}$ invasion and thermodynamic buffering effects are obtained by adding the preformed ΔDIC from Eq. 3 to a 7-year modeled average of historic DIC at each grid cell (Eq. 4). The 7-year average of modeled modern TA and the estimate of modern DIC from Eq. 4 inform estimates of $p\mathrm{CO_2}$ and pH.

$$\mathrm{DIC_{estimate,\,modern}} = \mathrm{DIC_{model,\,historic}} + \Delta\mathrm{DIC_{preformed}} \qquad (4)$$

## Data availability
The geochemical datasets generated in this study are provided in the Source Data file. Model analyses or model output data are available from the corresponding author on reasonable request.

## Code availability
The code used in this study is archived at: https://doi.org/10.5281/zenodo.16747884[55] and is also available at: https://github.com/mmstoll/WestCoast_IndustrialEra_OA. This repository includes Jupyter notebooks and Python scripts to reproduce the main figures and analyses presented in this manuscript.

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

## Acknowledgements

This project was supported by a grant from the Washington Ocean Acidification Center to A.C.G. We would like to thank the University of Washington Program on Climate Change and the Northwest Straits Foundation Caroline Gibson Scholarship for additional support and funding. Cruises funded by the Washington Ocean Acidification Center and NOAA's Ocean Acidification Program provided essential data for this study. We also thank the crew of the University of Washington R/V Rachel Carson for their support of a successful research cruise amid the COVID-19 pandemic. The modeling work was funded by grants from the National Science Foundation (OCE-1419323), the National Oceanic and Atmospheric Administration (DOC-NOAA NA15NOS4780186), and the Gordon and Betty Moore Foundation (GBMF #3775). Model simulations were carried out using the Yellowstone supercomputer supported by NCAR. S.R.A. thanks the NOAA Pacific Marine Environmental Laboratory for salary support. H.J. acknowledges funding from the Leverhulme Trust Early Career Fellowship (ECF-2023-199). Both J.W.B.R. and H.J. were supported by the European Research Council Horizon 2020 research and innovation program grant agreement number 805246. This is NOAA Pacific Marine Environmental Laboratory contribution number 5509 and CICOES contribution number 2024-1408.

## Author contributions

M.M.V.S. prepared coral samples for analysis, interpreted coral data and model simulations, co-organized the coral collection cruise, and prepared the manuscript under the supervision of A.C.G. and C.A.D. H.J. and J.W.B.R. measured and performed quality control on boron isotope data. H.F. performed ROMS simulations of the California Current System. S.R.A. provided oceanographic data from Washington Ocean Acidification Center cruises. A.C.G. and M.M.V.S. jointly conceived the study with A.M.G. and secured historic coral samples from the Smithsonian National Museum of Natural History. All authors edited the manuscript prior to submission.

## Competing interests

The authors declare no competing interests. Senior author A.C.G. is a co-founder and major shareholder of carbon removal company Banyu Carbon, Inc. This declaration is meant to avoid the appearance of a possible conflict of interest; however, this publication is unrelated to the work of Banyu Carbon. Banyu Carbon had no role in the research, and most of the research activities in the manuscript occurred before Banyu Carbon was formed.
