## [Peer Review file · Nature Communications]

A Century of Change in the California Current: Upwelling System Amplifies Acidification

Corresponding Author: Ms Mary Margaret Stoll

Version 1:

Reviewer comments:

Reviewer #2

(Remarks to the Author)

I commend the authors for combining paleo data with a modelling approach, which is a fresh and valuable contribution on a very relevant topic. However, I am concerned about the use and presentation of the historical paleo data. While the manuscript aims to compare present-day and historical paleorecords, the historical data seem relegated to the Extended Data and Supplementary Information, which diminishes their significance. The historical records are extensively discussed in the main text, yet the figures and some details are buried in the Extended Data and Supplementary Information. The Supplement should complement the main text, not serve as the primary presentation of critical results. This approach makes it difficult to fully appreciate the historical datasets, and the study appears overly reliant on the model data as a result. Why not show some of the historical data in figures 2 or 3?

The authors emphasize CO₂ as the focal point of their discussion, given its comparative utility across locations. However, CO₂ estimates depend on assumptions about DIC, which introduce additional layers of uncertainty. Moreover, the temperature effect on the k value is not incorporated into the calculation of pH from $\delta^{11}\text{B}$. Addressing these uncertainties explicitly would strengthen the study.

While the combination of paleo and modelling approaches holds promise, the manuscript would benefit from a clearer and more prominent presentation of the historical datasets. Improving transparency regarding uncertainties, sampling strategies, and data handling would greatly enhance the robustness of the study.

Specific Comments:

- Figure 3, panel e: The description of the presented data should clarify that it represents modelled data. Why are the historical coral data not included in this figure?
- Line 15: Consider revising “the leading edge” to “a leading edge,” depending on the context.
- Line 107: What about intraspecies variability, such as vital effects? This point needs clarification.
- Line 142: Specify explicitly that this refers to data from Gagnon et al. (2021).
- Line 609: The sampling strategy requires further detail. This concern was raised previously by reviewers, yet the revisions still lack sufficient information regarding how the corals were sampled. E.g., which skeletal elements were sampled, distance, etc.

Additionally, potential offsets in the data could result from treatment effects (e.g., preferential dissolution of skeletal elements like the centres of calcification) or sampling strategies that selectively represent certain skeletal components. In this sense, is there any information on how the historical corals were treated post-collection, such as cleaning or preservation procedures?

Furthermore, as indicated by the authors critical information is lacking for some historical samples, particularly sample 7863. Why was dating not performed on this sample? Providing this detail could help address gaps in the dataset.

(Remarks on code availability)

Reviewer #3

(Remarks to the Author)

Review: A Century of Change in the California Current: Upwelling System Amplifies Acidification

The current study investigates the speed of ocean acidification (OA) in the California Current System (CSS), a system where local waters are naturally acidified due to the upwelling of CO₂ enriched deep water. To estimate the pace of OA in the CSS compared to the global open ocean average, the authors rely in the first instance on the comparison of the $\delta^{11}\text{B}$ signal in the skeletons of deep-water coral (*Balanophyllia elegans*) which were collected recently and at the beginning of the industrial era. To identify which processes drive OA in the CSS and the Salish Sea the authors then use a regional ocean modelling system (ROMS) and a box model (Salish Sea). The authors find that the mean skeletal $\delta^{11}\text{B}$ signal of historic coral is significantly higher than that of modern coral which suggests a change over the last 100 years of $172 \pm 41 \mu\text{atm}$ regarding pCO₂ and -0.095 pH respectively. However, this change is not uniform, and OA appears to be especially amplified at depths below 50m. The two models generally support the geochemical record and suggest that the main drivers for the acceleration of OA in the CSS are the increase in atmospheric CO₂ as well as a thermodynamic buffering effect. The study sent to us for review has gone through an extensive peer review process already. We felt that the comments from the previous reviewers have been adequately addressed and that the study is overall well executed and the results credible. Thus, we have very little to criticize. Nonetheless, there are a few things that require further clarification in our opinion. Thus, we recommend the paper to undergo minor revisions before publication can be considered. Detailed comments below.

Comments:

The collection of the skeletal material for the $\delta^{11}\text{B}$ requires some further clarification. The authors write that 3 to 9 mg of material was collected from along the primary growth axis. So, does the collected/analyzed material represent the average over the whole growth period? How fast do these corals grow and how old do they get? The age of the historic coral at the time of collection is unlikely to affect the study outcome as pH values were, assumedly, relatively stable. However, if the modern coral are 100 years old the presented data would represent the average conditions over the whole period. Thus, if the authors would have considered only the most recently deposited material, then the pace of OA in the system may have been even faster.

Related to this, if the analyzed material indeed represents an average then this may, however, impact the depth-acidification patterns observed. If the populations in the different depth strata have a different age structure, then you would over or underestimate the pace of OA. For example, if the samples from the 75m stratum are the youngest then that would indeed suggest a faster pace of OA in that layer compared with the rest. Thus, I think it is important to provide some estimates of their respective age at these different depths.

L350 in the Supplementary Material: A change in river water discharge was discarded by the authors as the most likely cause for causing the observed patterns in acidification. The authors make this assumption on the data from the Skagit River Gauge using data from 1940–2020. Older data is then extrapolated. However, three major dams (Ross, Diablo and Gorge) of the Skagit River were completed or nearly completed before that (1924, 1936, and 1940). Thus, the extrapolation likely underestimates any changes in river discharge since they only incorporate data after major dams have been built. This requires some more research and justification from the authors.

Extended Data Figure 3: Looking at the observational data. It seems that they fall in between the mixing/upwelling line and CO₂ invasion/photosynthesis line suggesting an influence of both processes. Can the authors clarify how this would affect the calculation of their pH and pCO₂ trends over time and why this can be excluded?

Supplementary material L406-409: What is the reason for not having considered the input from the Skagit River? In the previous paragraphs the authors have highlighted the contribution of its water to the basin “variation in Salish Sea carbonate chemistry is strongly influenced by water mass mixing”, “The Skagit River represents the largest freshwater input to Puget Sound” and “Furthermore, normalizing the TA and DIC for salinity causes the data to collapse and a majority of the seasonal variability is removed”.

For their simulations it seems that the authors have selected to present their data on the seawater scale while the geochemical data return pH on the total scale. Since the authors are interested in comparing respective differences the outcome for the study is the same. Also, the choice of the authors for focusing on changes in pCO₂ also avoids some of the potential pitfalls there. However, could the authors confirm that there are not any differences introduced by selecting two different pH scales? I was also wondering whether it would not be better to also or solely present the modelled data on the total scale to make them match with the geochemical records and the discrete water samples. I gathered that these mostly report pH on the total scale.

Minor comments:

Fig1b: Isn't the CSS also a source of CO₂(atm)? The authors could consider adding an arrow going back to the atmosphere.

L241: Likely that we missed it but how was upwelling velocity calculated and what is the underlying data for it?

L643: Regarding the t-test for the coral samples and all other test performed. How were the test assumptions (e.g., normality) assessed?

Figure 2: It appears that the error bars are in some cases taken from the population mean. I am unfamiliar with this approach

and wonder how reliable these are? In any case, it is unclear to us with population(s) are the respective values taken from? I think the authors don't lose much removing the bars for those samples where this was done (n=1) if they decide to keep them than it should be clear where they come from.

L203: I find the word reproduce is a bit odd here. We'd argue that the coral record represents the more solid data, even though it is only proxy data, and thus the model is the one that reproduces patterns.

L709: I find the references used here slightly misleading. It sounds like they support the statement regarding elevated growth of *B. elegans* in summer but actually both references only make statements with regard to another cold-water coral (*D. dianthus*). The authors themselves write this out a bit more clearly in the Supplementary Material. I agree that some similarities can be inferred here but some caution must be taken with making these statements as we are dealing with a different species here and likely also different systems. Also, do the authors have more evidence to offer that summer conditions in the CSS could indeed facilitate growth of *B. elegans* in summer?

Extended data:

Extended Data Figures 7 & 8: Suggest using a different scale of colours to better differentiate the area of interest (especially when values are close to 1) from the background.

Supplementary material

L43-45: A reference should be added to support this concept. I suggest Chan 2017 (doi: <https://doi.org/10.1038/s41598-017-02777-y>) and/or Reum 2015 (doi: <https://doi.org/10.1093/icesjms/fsu231>).

L66-73: Some of these statements would benefit from a reference to back them up.

Figure S1.1 & S1.2: The resolution seems very low. Also, check title and axis labels for S1.2.

L204: Table description above the table.

L220: "0" missing for -.101 offset.

L344: The use of "Although" makes the sentence in contraposition with what has been said before, while it is not.

L346: Check 2nd citation of Kroeker et al. 2013. The citation got his own sentence there.

Line 487/Figure S3.2: See comment reading Figure 2 above.

Line 517: I think here a brief description of the methodology could be useful instead of only referring the reader to another paper.

(Remarks on code availability)

Reviewer #4

(Remarks to the Author)

(Remarks on code availability)

Version 2:

Reviewer comments:

Reviewer #2

(Remarks to the Author)

The revisions have improved the manuscript, adding greater clarity to the methodology. The conclusions appear to be well supported by the data, so I do not believe further revisions are necessary. Therefore, I am supportive of the publication of the current version of the manuscript. I have only two minor comments that can be addressed during the final preparation:

Line 433 – The ideas in this sentence are difficult to follow. Is "larger" a typo? Please revise for clarity.

Line 460 – The comment regarding sample resolution primarily referred to the older samples. Please clarify whether any information is available about how these samples were treated after collection (e.g., whether they were bleached).

(Remarks on code availability)

Reviewer #3

(Remarks to the Author)

The authors have satisfactorily addressed the previous comments and I can recommend the acceptance of the manuscript.

(Remarks on code availability)

I skimmed over the code but didnt run it. From what I could see, the code should be easy to run. It is accompanied by explanatory text and seems well structured.

Reviewer #4

(Remarks to the Author)

(Remarks on code availability)

Responses to Reviewer Comments

‘A Century of Change in the California Current: Upwelling System Amplifies Acidification’

We thank all referees for their thorough assessment of our work, careful reading of the manuscript, and numerous insightful comments and suggestions. These thoughtful reviews provided an opportunity to clarify important points and details concerning the interpretation and assumptions made in our analyses. Below please find a point-by-point response to all comments, along with a revised manuscript with tracked changes that reflect the constructive feedback provided in review.

REVIEWER COMMENTS

Reviewer #3 (Remarks to the Author):

“The current study investigates the speed of ocean acidification (OA) in the California Current System (CSS), a system where local waters are naturally acidified due to the upwelling of CO₂ enriched deep water. To estimate the pace of OA in the CSS compared to the global open ocean average, the authors rely in the first instance on the comparison of the $\delta^{11}\text{B}$ signal in the skeletons of deep-water coral (*Balanophyllia elegans*) which were collected recently and at the beginning of the industrial era. To identify which processes drive OA in the CSS and the Salish Sea the authors then use a regional ocean modelling system (ROMS) and a box model (Salish Sea). The authors find that the mean skeletal $\delta^{11}\text{B}$ signal of historic coral is significantly higher than that of modern coral which suggests a change over the last 100 years of $172 \pm 41 \mu\text{atm}$ regarding pCO₂ and -0.095 pH respectively. However, this change is not uniform, and OA appears to be especially amplified at depths below 50m. The two models generally support the geochemical record and suggest that the main drivers for the acceleration of OA in the CSS are the increase in atmospheric CO₂ as well as a thermodynamic buffering effect. The study send to us for review has gone through an extensive peer review process already. We felt that the comments from the previous reviewers have been adequately addressed and that the study is overall well executed and the results credible. Thus, we have very little to criticize. Nonetheless, there are a few things that require further clarification in our opinion. Thus, we recommend the paper to undergo minor revisions before publication can considered. Detailed comments below.”

Comments:

“The collection of the skeletal material for the $\delta^{11}\text{B}$ requires some further clarification. The authors write that 3 to 9 mg of material was collected from along the primary growth axis. So, does the collected/analyzed material represent the average over the whole growth period? How fast do these corals grow and how old to they get? The age of the historic coral at the time of collection is unlikely to affect the study outcome as pH values were, assumedly, relatively stable. However, if the modern coral are 100 years old the presented data would represent the average conditions over the whole period. Thus, if the authors would have considered only the most recently deposited material, then the pace of OA in the system may have been even faster.”

- We thank Reviewers #3 and #4 for their valuable insights and helpful suggestions. This comment provided the opportunity to clarify an important detail previously missing from the Methods section discussing boron isotope analyses and interpretation. These

clarifications have been added to our Methods section (lines 419-431 and 434-453), and we thank the reviewers for raising this point.

- The reviewers are correct that the collected/analyzed material represents an average over the entire growth period, however this is relatively short: it has been estimated that the average life span for *B. elegans* in the California Current is 6.5 to 11 years (Fadlallah 1983). As noted, the age of the historic coral at the time of collection is unlikely to affect the study outcome, as pH values were assumed to be relatively stable. The age of the modern corals is also unlikely to impact the modern pH values, as they represent conditions over a narrow time window (5-10 years). Given that the modern corals were collected in 2020, they reflect conditions from approximately 2010 to 2020. Any uncertainty in the exact age and therefore assumed pace of acidification over this period is within the measurement error of the corals.

“Related to this, if the analyzed material indeed represents an average then this may, however, impact the depth-acidification patterns observed. If the populations in the different depth strata have a different age structure, then you would over or underestimate the pace of OA. For example, if the samples from the 75m stratum are the youngest then that would indeed suggest a faster pace of OA in that layer compared with the rest. Thus, I think it is important to provide some estimates of their respective age at these different depths.”

- We thank the reviewers for carefully considering how these factors may influence the interpretation of coral data. We have added several sentences to the Methods section (Sample Collection, lines 419-431) to discuss coral ages and highlight the narrow time frame during which the historic corals from the CCS were collected.
- While we have not found any evidence to suggest that populations at different depth strata have different age structure, this effect is unlikely to impact our study results given that each coral represents conditions over such a narrow time window. Additionally, all of the historic corals in the CCS were collected within 1.5 years of each other (September 1888 to March 1890). The natural $p\text{CO}_2$ gradient with depth is also much larger than differences in the pace of OA over a 5-10 year time period.

“L350 in the Supplementary Material: A change in river water discharge was discarded by the authors as the most likely cause for causing the observed patterns in acidification. The authors make this assumption on the data from the Skagit River Gauge using data from 1940–2020. Older data is then extrapolated. However, three major dams (Ross, Diablo and Gorge) of the Skagit River were completed or nearly completed before that (1924, 1936, and 1940). Thus, the extrapolation likely underestimates any changes in river discharge since they only incorporate data after major dams have been built. This requires some more research and justification from the authors.”

- The reviewers raise an interesting concern regarding our assumptions about major dams on the Skagit River. We acknowledge that our extrapolation likely underestimates changes in river discharge, as the river gauge was put in place after major dam construction. To assess the implications for our acidification estimates and interpretation, we conducted additional research. A summary of this analysis is now included in Supplementary Discussion 1.1 (lines 428-456), with details provided below.
- First, we identified another USGS River Gauge (12194000) on the Skagit River that extends further back in time to 1925. We found that Gorge Dam, Diablo Dam, and Ross Dam were completed in 1924, 1931, and 1940, respectively (Skagit County Basin

Overview Report). This river discharge record includes data before the completion of the Diablo and Ross dams, but after the Gorge Dam was built. Among these, only Ross Dam is a storage dam, meaning it had the potential to significantly reduce natural spring river flows from snowmelt (Pacific International Engineering, 2008). The impact of the Ross Dam, which most likely influenced acidification the most as a large storage dam, is captured in the new river gauge data.

- We applied the same analysis as described in Supplementary Discussion 1.1 to evaluate whether a shift in mixing could have driven 20th century acidification. A best-fit linear regression was applied to USGS River Gauge (12194000), and we found that river discharge actually increased by ~5% between 1925–2020. However, this period includes the Dust Bowl, which caused prolonged droughts and several consecutive low-precipitation years between 1928 and 1942 (Washington State Governor’s Report). If we assume that the pre-drought years better represent historical conditions (after Gorge Dam was built), we estimate that river discharge declined by ~6%.
- We also consulted with the Army Corps of Engineers, who manage the Skagit River dams, but they unfortunately did not have records of river flow before Gorge Dam was built in 1924. However, given that Gorge Dam was not a storage dam, this absence of data is unlikely to affect our conclusions.
- Thus, while dam construction likely influenced carbonate chemistry in the Admiralty Inlet area, the corals conservatively suggest that river discharge would have had to decline by over 20% to explain the observed acidification. Although dam construction likely influenced carbonate chemistry to some extent in the Salish Sea, changes in river discharge still cannot account for the main driver of acidification.

“Extended Data Figure 3: Looking at the observational data. It seems that they fall in between the mixing/upwelling line and CO₂ invasion/photosynthesis line suggesting an influence of both processes. Can the authors clarify how this would affect the calculation of their pH and pCO₂ trends over time and why this can be excluded?”

- Changes in seawater chemistry over the past century are bounded by historic (green line) and modern (gray line) coral data in what is now Supplementary Fig. 4. Since the contour lines of constant $p\text{CO}_2$ (black dashed lines) and the coral-derived bounds on $p\text{CO}_2$ are roughly parallel, the coral data constrain $p\text{CO}_2$ tightly without additional process-based information. However, to better understand the dominant mechanism of acidification, we conducted additional analyses and process studies to explore potential drivers.
- The observational data in Supplementary Fig. 4 represent the seasonality of carbonate chemistry at the Washington Ocean Acidification Center (WOAC) station 21. The reviewer correctly notes that the data fall between the mixing/upwelling line and CO₂ invasion/photosynthesis lines, suggesting that all three processes influence seasonal carbonate chemistry in the Salish Sea. However, the observed pH and $p\text{CO}_2$ changes depend on assumptions about how these processes changed over the past century. To evaluate the dominant driver of 20th century acidification, Supplementary Discussion 1.1 presents process-based arguments for each mechanism shown in Supplementary Fig. 4. We find it is unlikely that changes in calcification, photosynthesis, or mixing ratios alone could explain the observed acidification.
- As a sensitivity test, we hypothesized that acidification over the past century was driven by a shift in the seasonality of carbonate chemistry in the Salish Sea. To estimate changes

in pH and $p\text{CO}_2$, we fit a best-fit line through the observational data (which accounts for all processes described above), and calculated its intersections with the historic and modern coral data. Under this assumption, $\Delta p\text{CO}_2 = 169 \text{ uatm}$, which is well within the uncertainty range of our estimate of $\Delta p\text{CO}_2 = 172 \text{ uatm}$ under the assumption of CO_2 invasion. In short, regardless of the assumed dominant process driving acidification, the overall magnitude of acidification is well constrained because $\delta^{11}\text{B}$ provides such a strong bound.

“Supplementary material L406-409: What is the reason for not having considered the input from the Skagit River? In the previous paragraphs the authors have highlighted the contribution of its water to the basin “variation in Salish Sea carbonate chemistry is strongly influenced by water mass mixing”, “The Skagit River represents the largest freshwater input to Puget Sound” and “Furthermore, normalizing the TA and DIC for salinity causes the data to collapse and a majority of the seasonal variability is removed”.”

- The purpose of this box model was to provide a simple estimate of acidification in the Salish Sea, independent of the coral records. To achieve this, we included only the dominant mechanism likely to drive acidification over the 20th century, as identified through the process-based arguments in Supplementary Discussion 1.1. Based on these arguments, we find that CO_2 invasion was the most likely primary driver.
- To evaluate acidification under this mechanism, we created a box model that estimates acidification solely driven by CO_2 invasion. Therefore, the model incorporates two key processes: estuarine circulation of upwelled waters from the CCS, which contain imprints of atmospheric CO_2 from when they were last at the surface, and air-sea gas exchange in the Salish Sea estuary.
- While the Skagit River plays an important role in the seasonality of carbonate chemistry in the Salish Sea, this box model focuses on long-term mean state changes over the 20th century. As discussed in Supplementary Discussion 1.1, we find that changes in river flow over the past century were unlikely to be the dominant driver of acidification.
- The close alignment between the coral-based and box model-based estimates of $\Delta p\text{CO}_2$ suggest that anthropogenic carbon, modulated by estuarine circulation and air-sea gas exchange, can account for the primary driver of 20th century acidification.

“For their simulations it seems that the authors have selected to present their data on the seawater scale while the geochemical data return pH on the total scale. Since the authors are interested in comparing respective differences the outcome for the study is the same. Also, the choice of the authors for focusing on changes in $p\text{CO}_2$ also avoids some of the potential pitfalls there. However, could the authors confirm that there are not any differences introduced by selecting two different pH scales? I was also wondering whether it would not be better to also or solely present the modelled data on the total scale to make them match with the geochemical records and the discrete water samples. I gathered that these mostly report pH on the total scale.”

- We thank the reviewers for highlighting this point regarding pH scales and the presentation of both modeled and geochemical data. The reviewers are correct that our focus on $p\text{CO}_2$ and respective differences between historic and modern eras avoids complications associated with using two pH scales. However, to maintain consistency, we updated what are now Figures 4a and 4c to show modeled data on the total pH scale. Changes to the plot are minimal; the difference between the total and seawater scales is typically on the order of 0.01 pH units, whereas the color bar in Figure 4a spans 0.3 units.

Minor Comments:

“Fig1b: Isn’t the CSS also a source of CO₂ (atm)? The authors could consider adding an arrow going back to the atmosphere.”

- The reviewers are correct that the CCS also acts as a source of CO₂ (atm) in some regions and seasons. We did actually consider adding an arrow near the upwelling region to show outgassing, but we decided to simplify the figure to emphasize processes relevant to the amplification effect.

“L241: Likely that we missed it but how was upwelling velocity calculated and what is the underlying data for it?”

- The upwelling velocities are extracted from the model output; details regarding the computation be found in Shchepetkin and McWilliams, 2008 (page 150).

“L643: Regarding the t-test for the coral samples and all other test performed. How were the test assumptions (e.g., normality) assessed?”

- Normality was evaluated for all statistical assessments using the Shapiro-Wilk test in python using the sci-kit learn package. The historic dataset from the West Coast and historic and modern datasets from the Salish Sea all appeared to be normally distributed.

“Figure 2: It appears that the error bars are in some cases taken from the population mean. I am unfamiliar with this approach and wonder how reliable these are? In any case, it is unclear to us with population(s) are the respective values taken from? I think the authors don’t lose much removing the bars for those samples where this was done (n=1) if they decide to keep them than it should be clear where they come from.”

- At locations where there was only one coral, we used the population standard deviation to represent the standard error of the mean. The population standard deviation was inferred by calculating the average of standard deviations at locations with 8+ samples, excluding USNM 7863 located in a highly variable tidepool environment. This method is described under Methods, ‘Coral-Based Estimates of Acidification in the CCS’; we added more detail to clarify where this value is derived. Brief information is included in the Figure 2 (now Figure 3) caption, but we also added a reference to the Methods section to provide more detail if the reader is interested (line 788-789).
- This is a conservative approach meant to indicate that there is also error associated with locations that have only one coral. The alternative of removing error bars for locations with only one coral could lead the reader to think that these locations are better known than they are. However, if the reviewers prefer we remove the error bars in Figure 3, we are open to doing so.

“L203: I find the word reproduce is a bit odd here. We’d argue that the coral record represents the more solid data, even though it is only proxy data, and thus the model is the one that reproduces patterns.”

- We thank the reviewers for pointing this out. We agree that the coral record represents more solid data and changed the phrasing of this sentence to reflect this perspective.

“L709: I find the references used here slightly misleading. It sounds like they support the statement regarding elevated growth of *B. elegans* in summer but actually both references only make statements with regard to another cold-water coral (*D. dianthus*). The authors themselves write this out a bit more clearly in the Supplementary Material. I agree that some similarities can be inferred here but some caution must be taken with making these statements as we are dealing with a different species here and likely also different systems. Also, do the authors have more evidence to offer that summer conditions in the CSS could indeed facilitate growth of *B. elegans* in summer?”

- Thank you for pointing out this missing information from the Methods section. We agree this is important information to include in the Methods for transparency, regardless of the detail included in the Supplementary Material. We added a statement to the Methods section (lines 663-665) to specify that these references describe growth patterns from a different but closely-related cold-water coral species.
- Further research provided additional evidence that summer conditions in the CCS could facilitate growth of *B. elegans*. Fadlallah 1983 discovered that food availability, as a result of upwelling, could promote growth and larval settlement. We included this additional reference in the Methods (lines 663-665) and Supplementary Methods 1.3 (lines 232-234) sections.

Extended Data:

“Extended Data Figures 7 & 8: Suggest using a different scale of colours to better differentiate the area of interest (especially when values are close to 1) from the background.”

- We agree that a different color bar would better differentiate the important areas in these figures (now Supplementary Fig. 9 and 10). We updated these figures with a new color bar.

Supplementary Material:

**We reordered the Supplementary Information sections and divided the text into Supplementary Methods and Discussion sections. We include updated line numbers for the relevant revisions below.

“L43-45: A reference should be added to support this concept. I suggest Chan 2017 (doi: <https://doi.org/10.1038/s41598-017-02777-y>) and/or Reum 2015 (doi: <https://doi.org/10.1093/icesjms/fsu231>).

- We agree that this statement is best supported by references to other studies. Thank you for providing these two articles – we included these references in lines 826-827.

“L66-73: Some of these statements would benefit from a reference to back them up.”

- We agree that these statements are best supported by references to back them up. We included additional references in lines 847-855.

“Figure S1.1 & S1.2: The resolution seems very low. Also, check title and axis labels for S1.2.”

- Thank you for pointing out these figure errors. We updated (now) Supplementary Fig. 14 with higher resolution and corrected the labels.

“L204: Table description above the table.”

- Thank you for catching this error. We moved the caption for Table S2.1 above the table.

“L220: “0” missing for -.101 offset.”

- Thank you for catching this error. We added a ‘0’ preceding the offset value.

“L344: The use of “Although” makes the sentence in contraposition with what has been said before, while it is not.”

- We can see how this statement may cause confusion, so we clarified this explanation in the Supplementary Discussion 1.1 (lines 393-401).

“L346: Check 2nd citation of Kroeker et al. 2013. The citation got his own sentence there.”

- Another great catch – thank you! We deleted the 2nd citation of Kroeker et al. 2013 in line 346.

“Line 487/Figure S3.2: See comment reading Figure 2 above.”

- Thank you for noting the error bars in Figure S3.2 as well – response is as for what was previously Figure 2, now Figure 3.

“Line 517: I think here a brief description of the methodology could be useful instead of only referring the reader to another paper.”

- I believe the reviewers here are referring to the Osborne et al. 2020 study, which reconstructed carbonate chemistry in the California Current using fossil foraminifera tests. The paragraphs between lines 690-707 provide a brief overview of the methodology used in Osborne et al. 2020 and describe how we utilized the data published in that study to compare to our coral record. We are unclear on the methodological scope that the reviewers are referring to, but we are happy to include additional detail with more direction.

Reviewer #4 (Remarks to the Author):

“I co-reviewed this manuscript with one of the reviewers who provided the listed reports. This is part of the Nature Communications initiative to facilitate training in peer review and to provide appropriate recognition for Early Career Researchers who co-review manuscripts.”

- We thank Reviewer #3 and #4 for their careful reading of the manuscript, and numerous insightful comments and suggestions.

Reviewer #2 (Remarks to the Author):

“I commend the authors for combining paleo data with a modelling approach, which is a fresh and valuable contribution on a very relevant topic. However, I am concerned about the use and presentation of the historical paleo data. While the manuscript aims to compare present-day and historical paleorecords, the historical data seem relegated to the Extended Data and Supplementary Information, which diminishes their significance. The historical records are extensively discussed in the main text, yet the figures and some details are buried in the Extended Data and Supplementary Information. The Supplement should complement the main text, not serve as the

primary presentation of critical results. This approach makes it difficult to fully appreciate the historical datasets, and the study appears overly reliant on the model data as a result. Why not show some of the historical data in figures 2 or 3?”

- We thank the reviewer for their thoughtful feedback and appreciation of our approach to integrate paleo data with modeling. We understand the concern regarding the placement of historical data and appreciate the opportunity for revision. Based on reviewer feedback, we embraced the core data of the study and moved what was previously Extended Data Figure 2 of Salish Sea paleo proxy data to the main text (now Figure 2). This figure shows the primary geochemical signal and that there is a significant difference between historic and modern corals collected from the same locations. Our efforts to better highlight geochemical data in the manuscript are discussed in more detail below.
- Salish Sea data:
 - The historic and modern coral records from the Salish Sea are now displayed in the main text as Figure 2. This geochemical data was distilled down to a single delta value (as discussed in Supplementary Methods 1.1) that captures the magnitude of acidification over the 20th century in terms of $p\text{CO}_2$ and pH. The significance of this value is thoroughly discussed in the main text, where we compare these changes to expectations if acidification had followed atmospheric trends. The observed changes in $p\text{CO}_2$ (+172 μatm) and pH (-0.095) from the coral records indicate that the Salish Sea experienced amplified acidification over the 20th century. This was the key takeaway from the Salish Sea geochemical data, and the added representation of this geochemical data in the main text emphasizes their importance to this study.
- West Coast data:
 - The historical West Coast coral data are presented in what is now Figure 3. The values on the y-axis for each data point represent historic $p\text{CO}_2$ averaged at each coral location. The $n = \#$ values next to each data point represent the number of corals that were averaged together at that location. The magnitude of acidification over the 20th century produced in the model is represented by the dashed lines to reflect different depth bins. This figure attempts to synthesize both geochemical and modeled data and emphasize how the magnitude of amplified acidification increases with depth in the CCS.
 - We chose to exclude the geochemical data from the West Coast (in what is now main text, Figure 2) because there is only historic data. Therefore, we thought our target audience would not find a direct, geochemical representation as intuitive without a comparison to modern data for context. The individual and location-averaged skeletal data from the West Coast can be found in Supplementary Fig. 2.
 - Figure 3e (now Figure 4e) shows modeled averages of seawater $p\text{CO}_2$ across the CCS at three depth levels and three time steps. We excluded the historic coral data because the corals were collected across a broad geographic and depth range with high variability in background seawater chemistry, making them unsuitable for basin-wide and depth-bin averaging. The variability across sites and depths can be seen by the spread in historic $p\text{CO}_2$ values along the y axis in Figure 3. Therefore, the coral data would not ‘bin’ adequately into the regional and depth bins of Figure 4.
- Supplementary figures:

- Supplementary Fig. 2 presents skeletal $\delta^{11}\text{B}$ data from individual corals and location averages from the Salish Sea and West Coast. Supplementary Fig. 2b is now reproduced in the main text to more prominently display the historic and modern geochemical data from the Salish Sea.
- We hope the above explanation clarifies our rationale for the balance between content in the main text and Supplementary Information. We are open to relocating additional content if the reviewer has specific suggestions.

“The authors emphasize CO_2 as the focal point of their discussion, given its comparative utility across locations. However, CO_2 estimates depend on assumptions about DIC, which introduce additional layers of uncertainty. ... Addressing these uncertainties explicitly would strengthen the study.”

- We thank the reviewers for pointing this out and appreciate the opportunity to clarify this important methodological detail. The main text (lines 129-136) now notes that the coral records provide a tight constraint on $p\text{CO}_2$ and references Supplementary Fig. 4. This revision includes an additional reference in the main text (line 136) to Methods (Coral-Based Estimates of $p\text{CO}_2$), where these assumptions and uncertainties are discussed. We also explicitly mention the robustness of $p\text{CO}_2$ estimates to ‘changes in DIC’ in line 135 to improve transparency. We also add a sentence to the Methods to explicitly state that the effect of DIC falls within measurement error of the corals (lines 562-564) and is not necessary to constrain our estimates of acidification (lines 568-571).
- One of the main points of the study is that $\delta^{11}\text{B}$ provides a strong constraint on $p\text{CO}_2$ regardless of the DIC value. To show this, we plot how a specific $\delta^{11}\text{B}$ value would be represented in pH vs DIC space in Supplementary Fig. 4. Historic and modern skeletal $\delta^{11}\text{B}$ measurement averages in the Salish Sea are indicated by solid lines. We found that $\delta^{11}\text{B}$ provides a strong constraint on $p\text{CO}_2$ regardless of the DIC value. Across a large range of DIC values (x-axis), lines of constant $\delta^{11}\text{B}$ are nearly parallel with the contour lines representing specific seawater $p\text{CO}_2$ values. This indicates that the corals provide a robust estimate of $p\text{CO}_2$ changes over the past century, even if DIC is not well known. While it is useful to have a rough idea of DIC for the general region of the ocean where the coral grew to best plot the right area of pH vs DIC space, this rough regional DIC does not need to be well known. The uncertainty in the selected DIC value to operationally calculate $\Delta p\text{CO}_2$ is well within measurement error and averaging error of the corals across locations.

“Moreover, the temperature effect on the k value is not incorporated into the calculation of pH from $\delta^{11}\text{B}$. Addressing these uncertainties explicitly would strengthen the study.”

- We now include several sentences to address this uncertainty in the Methods section ‘Coral-Based Estimates of $p\text{CO}_2$ in the Salish Sea’ in lines 622-636. We now show in this section (and describe below) that this is not a significant effect.
- We assessed the temperature effect on the k value by conducting a sensitivity analysis on our $p\text{CO}_2$ calculations. To calculate coral-based $p\text{CO}_2$ in the Salish Sea, we used a carbonate chemistry solver script in python with inputs including pH (from $\delta^{11}\text{B}$), DIC, temperature and salinity. Temperature values were based on summertime averages from bottle data collected at the Washington Ocean Acidification Center (WOAC) station 21, located near a majority of coral sampling sites.

- Under the assumption that temperature remained constant over the past century, we calculated $\Delta p\text{CO}_2 = 172$ uatm. However, if we assume that historical temperatures were 1°C cooler, then $\Delta p\text{CO}_2$ increases slightly to 176 uatm. This temperature effect is well within the uncertainty range of the coral records (+/- 41 uatm). Furthermore, if temperatures have indeed warmed over the past century, our acidification estimates would be even larger and more extreme. We adopt a conservative approach and assume a constant temperature over the past century, and thus report $\Delta p\text{CO}_2 = 172$ uatm in the Salish Sea.

“While the combination of paleo and modelling approaches holds promise, the manuscript would benefit from a clearer and more prominent presentation of the historical datasets. Improving transparency regarding uncertainties, sampling strategies, and data handling would greatly enhance the robustness of the study.”

- We thank the reviewer for recognizing our efforts to combine paleo and modeling approaches. Regarding the presentation of historical geochemical datasets, please refer to our response to the first comment above and the newly added Figure 2 to show geochemical data.
- In terms of sampling strategy, we see that Reviewer #3 also raised concerns about missing details. We agree that this description was lacking critical information and have now included additional clarity in the Methods section (lines 419-431, 434-453, and 463-481).
- We made a concerted effort to thoroughly document and explain our choices in data analysis and interpretation, including discussions on uncertainty and assumptions. We have addressed specific line by line comments as best as we can and hope these revisions enhance the clarity of the manuscript. We would be happy to provide additional explanation or expand the discussion further, if helpful.

Specific Comments:

“Figure 3, panel e: The description of the presented data should clarify that it represents modelled data. Why are the historical coral data not included in this figure?”

- We clarified in the figure caption that the data presented in Figure 4e represent modelled data.
- Figure 3e (now Figure 4e) shows modeled averages of seawater $p\text{CO}_2$ across the CCS at three depth levels and three time steps. We excluded the historic coral data because the corals were collected across a broad geographic and depth range with high variability in background seawater chemistry, making them unsuitable for basin-wide and depth-bin averaging. The variability across sites and depths can be seen by the spread in historic $p\text{CO}_2$ values along the y axis in Figure 3. Therefore, we don’t believe the coral data would ‘bin’ adequately into the regional and depth bins of Figure 4.

“Line 15: Consider revising “the leading edge” to “a leading edge,” depending on the context.”

- We agree with this suggestion and revised the sentence to say “a leading edge”.

“Line 107: What about intraspecies variability, such as vital effects? This point needs clarification.”

- We thank the reviewer for pointing out this information lacking in the Methods. The overall vital effects for this species are constrained by the calibration of Gagnon et al. (2021),

which is used here for the conversion of $\delta^{11}\text{B}$ to pH. This calibration, conducted with multiple individuals of corals of the same species, suggests that intraspecies variability is small compared to environmental effects. Indeed, the high precision of the Gagnon et al. calibrations and the striking similarity of the slopes for environmental sensitivity between Gagnon et al. and other cold-water coral studies support the utility of the $\delta^{11}\text{B}$ proxy. See for example Supplemental Figure 13 in Gagnon et al.

- Additionally, our comparison in the Salish Sea focuses on mean $\delta^{11}\text{B}$ values across modern and historic samples from the same locations. This approach helps mitigate the influence of individual variability and highlights broader temporal trends. Any potential intraspecies differences would be present in both modern and historical datasets, making it unlikely that vital effects alone explain the observed shift in mean $\delta^{11}\text{B}$.
- A summary of this information is now included in the Methods, Boron Isotope Analysis in lines 436-453.

“Line 142: Specify explicitly that this refers to data from Gagnon et al. (2021).”

- Thank you for this suggestion to improve clarity. We now explicitly reference Gagnon et al. 2021 in our discussion of culture conditions used to establish the $\delta^{11}\text{B}$ calibration.

“Line 609: The sampling strategy requires further detail. This concern was raised previously by reviewers, yet the revisions still lack sufficient information regarding how the corals were sampled. E.g., which skeletal elements were sampled, distance, etc.”

- Thank you for describing specific information that is requested regarding the sampling scheme. We included additional information of how the individual corals were sampled for geochemical analyses in Methods, ‘Boron Isotope Analysis’ in lines 436-453.
- Skeletal sections of ~5 mg CaCO_3 were sampled along the primary growth axis from each coral using a Dremel tool approximately ~0.5-1 cm deep. Sampling is large compared to skeletal microstructures like centers of calcification, which we think means that the impacts of these features are largely averaged and homogenized across the bulk sample. This is supported by the systematic trends and tight precision of even smaller samples from Gagnon et al. (2021).

“Additionally, potential offsets in the data could result from treatment effects (e.g., preferential dissolution of skeletal elements like the centres of calcification) or sampling strategies that selectively represent certain skeletal components. In this sense, is there any information on how the historical corals were treated post-collection, such as cleaning or preservation procedures?”

- As mentioned below and added to the text in lines 401-403, 419-431, and 436-453, samples incorporate a representative average of coral skeletal components by sampling broadly along the full growth axis. This same sampling strategy was employed for all corals, so should not influence the signals seen in our dataset. Care was also taken during cleaning to treat all samples identically and thus avoid preferential dissolution of specific components.
- All of the coral samples from the Smithsonian National Museum of Natural History were dried for archive after removing the coral polyp, avoiding potential issues around preferential dissolution that could arise from storage in solutions. Calcium carbonate is relatively stable in dry environments, and the museum employs preservation methods to ensure skeletal material remains intact. The Smithsonian implements preventative

measures, including environmental regulation (ie temperature and humidity) and archive handling, to maximize the long-term preservation of specimens.

- Smithsonian Preservation Guidelines:
 - <https://siarchives.si.edu/what-we-do/preservation#:~:text=The%20first%20and%20most%20important,the%20best%20rule%20of%20thumb.>

“Furthermore, as indicated by the authors, critical information is lacking for some historical samples, particularly sample 7863. Why was dating not performed on this sample? Providing this detail could help address gaps in the dataset.”

- We add additional detail on these choices in Supplementary Methods 1.4 in lines 313-319, and also describe this additional detail below.
- USNM 7863 was the only coral sample set lacking a recorded collection date. To address this gap, we conducted research on the known collector, Ernest Quayle, to provide transparency and infer unknown information (information included in Supplementary Methods 1.4). Quayle moved to California in 1931, collected other corals in our dataset in 1932, and published an article on corals that same year. Based on this timeline, we infer that USNM 78638, collected near San Francisco, California, was likely obtained in the early 1930s. While Quayle passed away in the 1950s, providing an upper bound on collection timing, a later collection date would not substantially impact the coral data, as the majority of anthropogenic CO₂ emissions occurred later in the 20th century.
- Perhaps most notably, these corals were collected from a tidepool near Moss Beach, California—a highly variable environment. This was the only location where modern *p*CO₂ appeared lower than historic *p*CO₂, likely due to the extreme variability of the intertidal zone that influenced the $\delta^{11}\text{B}$ skeletal signatures. Given this high natural variability, long-term signals can be obscured by environmental noise. Consequently, these data were not heavily weighted in our analysis or conclusions. Had they been central to our study, additional efforts to constrain collection dates and reduce uncertainty would have been undertaken.